# Weed Management in Edamame Soybean Production

**DOI:** 10.3390/plants14223438

**Published:** 2025-11-10

**Authors:** Natalija Pavlović, Željko Dolijanović, Milena Simić, Vesna Dragičević, Miodrag Tolimir, Margarita S. Dodevska, Milan Brankov

**Affiliations:** 1Maize Research Institute “Zemun Polje”, 11185 Belgrade, Serbia; npavlovic@mrizp.rs (N.P.); smilena@mrizp.rs (M.S.); vdragicevic@mrizp.rs (V.D.); mtolimir@mrizp.rs (M.T.); 2Faculty of Agriculture, University of Belgrade, 11080 Belgrade, Serbia; dolijan@agrif.bg.ac.rs; 3The Institute of Public Health of Serbia “Dr Milan Jovanovic Batut”, 11000 Belgrade, Serbia; margarita_dodevska@batut.org.rs

**Keywords:** vegetable soybean, herbicide efficacy, pod yield, variety competitiveness, meteorological conditions, weed biomass, weed control

## Abstract

Weeds are among the primary constraints reducing soybean productivity, and their effective control is especially important in edamame, a vegetable soybean valued for its nutritional potential. As chemical control remains the dominant strategy, rational herbicide use is essential. This study aimed to evaluate the response of two edamame varieties (Chiba Green and Midori Giant) and the effectiveness of applied herbicides in weed control during the 2022–2024 growing seasons. Treatments included the following: pre-emergence herbicides (*S*-metolachlor + metribuzin) (H1); pre- (*S*-metolachlor + metribuzin) and post-emergence herbicides (imazamox + cycloxydim) (H2); and an untreated control (H0). The growing season influenced pod yield and biomass, with the highest yield recorded in 2022 (11.7 t ha^−1^), while variety affected only pod yield: on average, Midori Giant outperformed Chiba Green (10.6 vs. 6.1 t ha^−1^). Herbicide treatment affected weed dry biomass (3.3 g m^−2^ in H2 compared to 341.8 g m^−2^ in H0) and pod yield (4.3 t ha^−1^ in H0 for Chiba Green compared to 11.9 t ha^−1^ in H2 for Midori Giant). The results indicate that pre-emergence herbicides could satisfactorily reduce weed infestation under suitable meteorological conditions. The combined application of pre- and post-emergence herbicides increases production security (particularly in seasons with higher weed infestation), likely by extending the weed control period through pre- and post-emergence herbicide combinations, targeting different weed species during the soybean vegetative period. In addition, weed diversity was associated with a yield increase in Midori Giant. This research provides practical information and options for weed management in edamame production in the Western Balkan region.

## 1. Introduction

Soybean (*Glycine max* (L.) Merr.) is the third most widely grown crop in Serbia, after maize (*Zea mays* L.) and wheat (*Triticum aestivum* L.), covering approximately 211,020 ha of harvested area [1]. Apart from Russia and Ukraine, Serbia was, after Italy, the leading soybean producer in Europe in 2023 [1]. Moreover, it is the largest exporter of non-genetically modified (non-GM) soybean in southeast Europe [2]. Over a 10-year period (2013–2023), harvested areas under soybean slightly increased [1], due to harmonization with European trends to provide a domestic plant-based protein source and reduce dependence on overseas imports. However, a 10.3% decline between 2022 and 2023 suggests potential shifts in production dynamics [1].

In addition to its importance for food and feed production, soybean has immense value in crop rotation and soil fertility improvement, due to its nitrogen-fixing ability and contribution to soil nitrogen enrichment [3]. Soybean is highly sensitive to drought during germination, flowering, pod and seed formation, and seed filling [4]. The occurrence of water deficit accompanied by high temperatures during reproductive stages can lead to significant yield reduction [5]. Thus, climate change, with increased drought frequency and heat waves, could thereby affect soybean production [6]. Developing soybean varieties that can maintain productivity and adaptability across varying environmental conditions is becoming increasingly important [7]. Moreover, additional losses are often caused by weed pressure. As a C3 plant, soybean exhibits lower photosynthetic efficiency and a slower early growth rate than many C4 weed species; therefore special attention should be given to weed management. As a wide-row crop, competition starts with weed emergence; thus the first six weeks are crucial for weed control in soybean [8].

In early spring, the most dominant dicotyledonous weed species in soybean crops in Serbia are *Amaranthus retroflexus* L., *Chenopodium album* L., *Convolvulus arvensis* L., *Cirsium arvense* (L.) Scop., *Datura stramonium* L., *Hibiscus trionum* L., *Sinapis arvensis* L., and *Polygonum aviculare* L., while dominant monocotyledonous weeds include *Panicum crus-galli* L., *Cynodon dactylon* (L.) Pers., *Setaria viridis* (L.) P.Beauv., and *Digitaria sanguinalis* (L.) Scop. [9]. Other significant species observed in soybean across the growing season include *Sorghum halepense* (L.) Pers., *Abutilon theophrasti* Medik., *Ambrosia artemisiifolia* L., *Polygonum lapathifolium* L., and *Xanthium strumarium* L. [9]. Weeds can cause up to 33% yield loss in soybean [10]. While the standard soybean is mostly grown and studied in Serbia, there is no evidence of research conducted on edamame soybean so far, including in the Western Balkan region. The main difference between edamame and standard grain-type soybean is the harvesting time. While standard soybean is harvested at full maturity (BBCH 89), edamame, as a vegetable type of soybean, is harvested at the BBCH 76 growth stage, when pods and beans are still green, and it is intended for human consumption. Edamame has a shorter vegetation period and larger seeds, so the sowing distance in a row is greater, while sowing density is lower [11]. Edamame was domesticated in China, and this country remains the main world exporter of edamame [12]. In the USA, edamame has been widely consumed, while in Europe it has been gaining popularity, as confirmed by the increasing number of studies focusing on its cultivation in recent years [13,14,15]. The share of edamame in global soybean production is close to 2% [16].

Information about weed management in edamame soybean is limited and has mostly been addressed through various approaches, including cultural practices such as cover cropping [17], tillage systems [18], and the consideration of the impact of seed size on weed competitiveness [17]; biological control using allelopathic plant extracts [19]; and chemical control through pre-planting, pre- and post-emergence herbicide applications [20,21,22,23,24,25]. Although no studies have specifically compared the early-season competitive ability of edamame and conventional soybean in the region, it is generally recognized that crops established at lower planting densities reach the canopy closure more slowly and are therefore less competitive with early-emerging, aggressive weeds [26]. Combined with the limited number of registered herbicides for this crop, these traits can lead to more severe weed problems if not properly managed [27,28].

While there is no information about edamame cultivation in the southern parts of Europe, including the Western Balkan region, where fluctuations in meteorological conditions driven by climate change could affect vegetable production, it could be predicted that edamame will gain popularity and occupy significant agricultural areas. Therefore, this research presents the first report on weed management in edamame in this region. It aims to provide practical recommendations for edamame production in southeast Europe by evaluating the effects of pre- and post-emergence herbicide treatments, variability in growing season, and cultivar on weed control and crop productivity.

## 2. Results

### 2.1. Impact of Year, Variety, and Herbicide Treatments on Weed Biomass, Edamame Biomass, and Pod Yield

The growing season did not influence weed fresh and dry biomass, but it did affect pod yield, as well as fresh and dry soybean biomass in both evaluated phases: phase I and phase II. The greatest average pod yield was achieved in 2022 (10.2 t ha^−1^), while in 2023 and 2024, it was reduced by 36% (8.2 t ha^−1^) and 50.4% (6.6 t ha^−1^), respectively. In 2022, edamame exhibited the highest average biomass accumulation, with a 7.9-fold increase in fresh mass (from 125.2 g m^−2^ in phase I to 994.4 g m^−2^ in phase II) and a 9.5-fold increase in dry mass from phase I (23.2 g m^−2^) to phase II (219.3 g m^−2^), whereas in 2023, the increases were 3.8-fold for fresh biomass (from 148.6 g m^−2^ in phase I to 558.7 g m^−2^ in phase II) and 4.7-fold for dry biomass (from 24.6 g m^−2^ in phase I to 114.4 g m^−2^ phase II); in 2024, the increases were 3.0-fold (from 75.1 g m^−2^ in phase I to 221.2 g m^−2^ in phase II) and 5.1-fold (from 16.5 g m^−2^ in phase I to 51.2 g m^−2^ in phase II), respectively, for fresh and dry biomass (Table 1).

Herbicide treatments affected weed biomass and pod yield. Weed dry mass averaged 3.3 g m^−2^ in the pre- and post-emergence treatment (H2), 126 g m^−2^ in the pre-emergence treatment (H1), and 341.8 g m^−2^ in the untreated control (H0). The highest weed fresh and dry biomass was recorded in the untreated control plots (2104.6 g m^−2^ and 341.8 g m^−2^, respectively), while the greatest weed biomass reduction occurred in the H2 treatment (167-fold for fresh weed biomass and 104-fold for weed dry biomass) compared to H0. Pod yield followed a similar trend, with the lowest value in H0, which was 1.8 t ha^−1^ lower than that in H1, while H1 yielded 1.2 t ha^−1^ less than H2. Accordingly, the highest average yield was achieved in the H2 treatment (9.7 t ha^−1^). While variety did not have an impact on weed biomass, it influenced pod yield. Thus, Midori Giant (V2) had a 4.5 t ha^−1^ higher pod yield than Chiba Green (V1), on average. Despite the variations in yield, the biomass values of these two varieties were similar in both phases.

When the interaction between variety and herbicide treatment was considered, differences were observed in weed dry biomass and density. The lowest values were recorded in the H2 treatment and the highest in the H0 treatment for both varieties (Figure 1). Interaction between variety and herbicide did not have an impact on soybean dry mass in either phase. However, edamame dry biomass was increased at V2, up to 5.9-fold in the H0 treatment and 7.3-fold in the H1 treatment, from phase I to phase II. At V1, the increase in biomass from phase I to phase II was greater in the control plots (H0), compared to the plots treated with pre-emergence herbicides only (H1) (8.8-fold vs. 7.5-fold, respectively). Although the differences in pod yield between H2 and H1 were not significant, the highest yields were achieved under the H2 treatment in both varieties, with V2 showing a 37% advantage over V1 (11.9 t ha^−1^ vs. 7.5 t ha^−1^).

### 2.2. Weed Biomass and Herbicide Efficacy

The highest weed infestation was recorded in 2022, as evidenced by the total weed biomass across all herbicide treatments (Figure 2). Weed biomass in control plots at V2 was nearly double that at V1 (3451.0 g m^−2^ vs. 1849.9 g m^−2^). In 2022, herbicide efficacy in the H1 treatment was 14.2% lower at V2 than at V1. Considering 2023, the control plots had total weed biomass levels slightly lower than those in 2022. However, weed biomass at V1 was nearly double higher compared to V2 (3273.05 g m^−2^ vs. 1705.24 g m^−2^). In 2023, after pre-emergence herbicide application, plots were almost weed-free, with an average efficacy of 96%. An unusual scenario was observed in 2024 at V1: weed fresh biomass in the H1 treatment was higher than that in the control plots, resulting in a complete lack of herbicide efficacy. Also, the efficacy of the H1 treatment at V2 was notably low (22.9%). Weed dry mass followed a similar pattern as fresh mass. The differences among all three herbicide treatments in weed fresh and dry biomass were noticed in each growing season, in both varieties, apart from V2 in 2023 with no significant differences between the H1 and H2 treatments.

### 2.3. Weed Diversity

The highest biodiversity index (H) values were observed at V2 in the H1 treatment in 2022 (1.62) and at V2 in H0 in 2023 (1.60). Aside from the consistently low weed diversity in the H2 treatment (0–0.17), the lowest value was recorded in V2 in H1 in 2024 (0.34) (Table 2). Weed diversity, expressed by the Shannon index, was generally higher in the control plots compared to H1, except in the 2022 growing season when the H’ value in H1 exceeded that in H0 for both varieties. In H2, weed diversity was almost completely absent.

Variations in weed diversity were presented across years. In the V1H0 treatment, diversity decreased from 2022 to 2023 but reached its highest level in 2024. In V1H1, diversity peaked in 2022 and then declined over the following two seasons. In V2H0, weed diversity was greater in 2023 compared to 2022, but it was at its lowest level in 2024. In V2H1, no weeds were recorded in 2023, whereas the highest diversity occurred in the first growing season.

### 2.4. PCA for Weed Biomass Productivity in Relation to Applied Herbicide Treatments and Edamame Variety

To evaluate the linkage between edamame varieties, applied treatments, and weed biomass productivity across the identified weed species, principal component analysis (PCA) was performed (Figure 3). The PCA showed that the three-component model explained 94.7% of the total variability. The first axis contributed 59.1% to the total variability and was positively correlated with *S. halepense*, *A. retroflexus*, *Amaranthus hybridus* L., *D. stramonium*, and *H. trionum*. The second axis contributed 24.8% to the total variability and showed a positive correlation with *S. nigrum* L., *A. theophrasti*, and *S. viridis,* while a negative correlation was observed with *C. arvensis*. The third axis contributed 10.8% to the total variability and was positively correlated with *C. album*, *Chenopodium hybridum* L., and *Portulaca oleracea* L. *C. album*, *S. nigrum*, and *A. theophrasti* varied mostly in V1H1, and *S. nigrum* and *A. theophrasti* also varied in the V2H1 combination (Figure 1). V1H0 contributed mostly to the variability in *C. album* and *P. oleracea*, while V2H2 and V1H2 contributed to the variability in *C. arvensis*. V2H0 mainly caused the greater variability in *S. halepense*, *D. stramonium*, *A. retroflexus*, *A. hybridus*, and *H. trionum*.

## 3. Discussion

### 3.1. Impact of Year, Herbicide Treatments, and Variety on Edamame Productivity

Owing to the increasing popularity of edamame soybean as a food and its specific growing demands as a vegetable crop, special attention should be given to the development of an appropriate technology, i.e., a combination of cropping practices adjusted to particular agro-climatic regions. From this standpoint, weed management plays an important role, when considering edamame development, biomass production, and especially yield potential. Herbicide use remains the most effective practice for highly demanding crops, such as edamame soybean [27]. Due to its sensitivity as a vegetable crop, yield and biomass accumulation varied across the growing seasons, with the highest values recorded in 2022, while 2024 proved to be the least favourable, despite irrigation being applied in both seasons. Although precipitation was scarce or poorly distributed in 2022, the deficit was mitigated by irrigation. Nevertheless, in 2024, the year with the lowest yield recorded, irrigation was not a presumable factor in the yield reduction observed. It should be noted that higher average temperatures occurred in July 2023 and 2024, during the most critical periods for pods formation and seed filling. It is well known that soybean responds to high temperatures with flower abortion and reduced grain filling, thereby decreasing yield [29]. Under Serbian agro-climatic conditions, soybean requires 300–350 mm of water during its critical growth stages [30]. Early development stages benefited from sufficient rainfall in 2023, whereas a dry period in June and early July overlapped with the critical seed-filling stage. Therefore, in 2023, precipitation reached the threshold; however its distribution was uneven. The least favourable conditions occurred in 2024, characterized by irregular precipitation patterns (with the majority of rainfall occurring on only a few days) and extreme heat (lasting day and night for 15 days), which adversely affected reproductive development.

Herbicide treatment had a significant impact on pod yield by reducing weed infestation, with the greatest values observed under the H2 treatment, where both pre- and post-emergence herbicides were applied. The results from the field trial in Arkansas, USA, showed a similar trend, with edamame seed yield being 2.3–4.3-fold higher with pre- and post-emergence herbicide treatments compared to the untreated control [22]. Both varieties exhibited similar responses to herbicide treatments, with biomass variation driven primarily by variety and treatment, rather than their interaction.

Variety influenced only pod yield. Midori Giant, as a variety with a longer vegetative period and greater biomass production, had a significantly higher yield than Chiba Green. Williams [27] compared edamame cultivars from different maturity groups and plant canopy characteristics, noting that the late-maturing cultivar, with greater height and a denser canopy, also had a higher yield. There was no difference in pod yield between the H1 and H2 treatments for both varieties. Although Midori Giant (V2) and Chiba Green (V1) are classified within the same maturity group [13,31], our observations indicate that V2 required approximately 10 additional days to reach the BBCH 76 stage (full seed development) compared to V1. According to Jankauskienė et al. [13], no differences were observed between Midori Giant and Chiba Green in pod yield. Both studies [13,31] also reported that Midori Giant is taller than Chiba Green, which may have contributed to the greater weed competitiveness of Midori compared to Chiba. On the other hand, Ogles et al. [31] described Chiba Green as the higher-yielding variety, indicating that yield performance may vary depending on environmental conditions.

Despite higher weed pressure and lower herbicide efficacy in the H1 treatment in 2022, V2 outperformed V1 in pod yield capacity. This raises questions regarding the varietal adaptability and relative contribution of environmental factors, including weed infestation level, in determining edamame yield.

### 3.2. Impact of Year, Herbicide Treatments, and Variety on Weeds

In contrast to the soybean productivity parameters, the growing season did not have an impact on weeds, which can be explained by the greater adaptability of weeds to variable weather conditions [32]. However, herbicide treatment had the greatest impact on weeds, with the highest weed biomass reduction achieved under the H2 treatment. Accordingly, the greater H’ values in control plots were expected, as the intensive use of chemical inputs is known to reduce weed diversity [33]. It should be emphasized that fluctuations in weed diversity induced by year highlight the influence of weather conditions on weed dynamics. In the least favourable growing season (2024), weed diversity in control plots reached its maximum in V1, whereas in V2 it dropped to the lowest level. This could be due to the reduced competitiveness of V1 (which had lower yield potential than V2) under the unfavourable weather conditions, allowing weeds to spread, while V2 appears to be more competitive and suppress weed diversity more effectively.

Herbicide efficacy often fluctuates due to unpredictable variations in environmental factors [34]. Accordingly, the highest weed presence was observed in 2022, when favourable weather conditions likely ensured adequate resources and use of environmental services not only for weeds but primarily for crop growth, minimizing the impact of weed competition on the crop. In contrast, the reduced herbicide efficacy observed in 2024, and particularly the greater weed infestation in H1 compared to the untreated control in this year, may be explained by extremely high temperatures and drought, which mainly affected crop response and growth, including reduced stomatal conductance and accelerated phenological development [35,36,37], resulting in low pod yield potential. While higher temperatures increase herbicide absorption and translocation, they also enhance their further metabolization, thus reducing herbicide efficacy [38] and potentially causing phytotoxic effects on crops. Soil conditions are also an important factor when considering herbicide efficacy; higher soil temperatures increase the rate of herbicide degradation due to increased microbial activity and chemical degradation. In addition, the pre-emergence herbicide in the H1 treatment likely suppressed less competitive weed species, creating more space and enabling late-emerging and more tolerant weed species with greater biomass productivity to occupy the free space [39]. Furthermore, the application of pre-emergence herbicides can cause shifts in weed flora characteristics as reflected by changes in species composition and diversity [40]. At the same time, the lowest weed biomass was recorded in 2024 compared to the other experimental years. Taking into account the greater adaptability potential of weeds and the harsh meteorological conditions, herbicide effectiveness was reduced. Meanwhile, in 2023, pre-emergence herbicide application proved successful, suggesting that weather conditions, particularly sufficient precipitation following herbicide application, enhanced herbicide activation and efficacy, so post-emergence treatment could be omitted. This aligns with findings by Meyer (2023) [41], who stated that the effectiveness of pre-emergence herbicides is highly dependent on soil moisture availability after treatment, as this transfers the herbicide into the weed seed zone and enhances herbicide uptake by roots [38]. It should also be mentioned that herbicide efficacy depends on weed water status, as water-stressed weeds will poorly absorb and translocate systemic herbicides.

Contrary to our findings, which indicated that cultivar had no significant effect on weed biomass, Williams [27] reported a significant influence of edamame variety on weed fresh biomass. This may be explained by the fact that their study included both determinate and indeterminate cultivars, whereas both cultivars in our experiment were determinate, which may differ in their response to weed competition. Khan et al. [42] evaluated various vegetative and yield parameters of determinate and indeterminate soybean lines and found that indeterminate lines exhibited greater leaf area and plant height. Such characteristics may be indicative of enhanced canopy development and, consequently, a higher potential for the weed suppression of indeterminate lines. It should also be noted that, irrespective of the insignificant differences in biomass formed by the two edamame varieties, greater variability in the biomass of various weed species was observed in V2 (Midori Giant). This suggests that morphological traits, such as canopy development [43,44], may influence weed infestation and especially weed diversity. Such information could be valuable for seed producers, when selecting a desirable variety, particularly for organic production. This finding aligns with the research of Ferrero et al. [45] who reported that greater soybean yields can be achieved in the presence of more diverse weed communities.

When examining the interaction between variety and herbicide treatment, weed diversity gradually decreased from 2022 to 2024 in V1H1. In V2H1, diversity was also the highest in 2022, while no weeds were observed in 2023, indicating that herbicide efficacy strongly depended on weather conditions. Nevertheless, weed dry biomass and density differed significantly among the H0, H1, and H2 treatments in both varieties. The weight of fresh weed biomass in 2022 and 2024 indicated that the application of pre-emergence herbicides alone could not be sufficient to control weeds, in regard to the combined application of pre- and post-emergence herbicides. Accordingly, the greater weed diversity observed in the H1 treatment suggests the lower effectiveness of the pre-emergence herbicides, likely due to their limited activity against perennial weeds [46]. Contrarily, in 2023, herbicide efficacy in H1 was sufficiently high to allow for the omission of an additional post-emergence application, particularly in V2. The obtained results are consistent with Pornprom et al. [20] who pointed out that pre-emergence herbicide application (a.i. metribuzin) was sufficient to control weeds and ensure the highest edamame yield (11.8 t ha^−1^) compared to manual weeding and untreated control plots. In line with the above, greater variability in weed dry biomass, particularly of *S. halepense*, *D. stramonium*, *A. retroflexus*, *A. hybridus*, *H. trionum,* and *C. arvensis*, was observed in the untreated control (H0). This indicates that, irrespective of edamame variety, herbicide treatments can reduce not only weed biomass but also weed diversity [47,48]. This effect is particularly evident in the H2 treatment in both varieties, which effectively reduced weed diversity and had a primary impact on the biomass variability in *C. arvensis*. Taking into account the greater weed biomass productivity in 2022 and 2024 (as opposite years from the meteorological viewpoint), together with the greater weed diversity and higher pod yield potential of V2, it could be supposed that Midori Giant, with its greater height and denser canopy [13,31], could suppress dominant weeds through resource competition. Consequently, this canopy structure likely supports the capacity of V2 to maintain higher yield when greater weed diversification was present, enhancing its overall competitiveness [45] in regard to Chiba Green.

The selected soybean varieties are organically grown and certified for organic production by the U.S. Department of Agriculture’s (USDA) National Organic Program. However, they might be cultivated under both organic and conventional production systems. In Europe, all herbicides are registered for use in a certain crop, in contrast to the US, where herbicide-ready crops are mainly cultivated [49]. Furthermore, based on herbicide selection in our study, soybean does not have absolute tolerance to metribuzin and *S*-metolachlor; its response depends strongly on soil type and environmental conditions, and sensitive varieties may experience injury [50]. While cycloxydim affects only annual grasses, herbicide imazamox is selective due to metabolic degradation in soybean plants [51,52].

According to the results present in this study, high-yield edamame production relies on herbicide application, owing to successful weed management, particularly in temperate regions. Vegetables, mostly leafy greens, are recognized as one of the main entry points of pesticides residues into the human diet [53,54], and edamame, being a vegetable, should also be considered within this context. Although several herbicide active ingredients have been registered as safe for edamame production in the United States [27,55], evidence on the actual residue levels in this crop remains limited. The scarcity of residue-focused studies, together with factors including varietal diversity, the mode of action of active ingredients, application methods, farmers’ practices, and region-specific environmental conditions, highlights the need for further research into pesticide residues in edamame.

## 4. Materials and Methods

### 4.1. Experimental Site

This experiment was conducted at the experimental field of the Maize Research Institute “Zemun Polje” (44°52′ N 20°20′ E), in Belgrade, Serbia, during the 2022–2024 growing seasons. Two edamame varieties, Chiba Green (V1) and Midori Giant (V2) (Wannamaker Seeds, Saluda, NC, USA), were used in this study. The selected varieties originated from organic production (certified by the seller) and are not genetically modified, nor do they possess any herbicide trait introduced through the breeding process (e.g., herbicide-tolerant cultivars). Sowing was performed with an arrangement of a 50 cm inter-row space and 7.5 cm intra-row space (266,000 plants ha^−1^). The size of the elementary plot was 20 m^2^, consisting of six soybean rows each 5 m long. This experiment was designed as a split plot, with four replications. The previous crop in each season was winter wheat (*Triticum aestivum* L.), and the applied herbicides were iodosulfuron-methyl-sodium + amidosulfuron and metsulfuron-methyl. In 2022 and 2024, sowing was carried out at the beginning of the third decade of April, while in 2023, it was performed at the beginning of May (Table 3). NPK fertilizer (6:24:12) was incorporated into the soil in the autumn in the amount of 250 kg ha^−1^, while 270 kg ha^−1^ of urea (46% N) was applied in the spring prior to edamame sowing (Table 3). Soil moisture measurements were taken at 15-day intervals, and irrigation was applied as needed to maintain 70% of the field capacity. The dominant weed species were as follows: *S. halepense*, *C. album*, *C. hybridum*, and *S. nigrum*.

### 4.2. Herbicide Treatments

The herbicide treatments included the following: H1—pre-emergence herbicide mixture (*S*-metolachlor + metribuzin); H2—a combination of the pre-emergence mixture (*S*-metolachlor + metribuzin) and post-emergence herbicides (imazamox and cycloxydim); H0—untreated control (Table 4). Pre-emergence herbicide treatment was carried out 1–2 days after edamame sowing. Post-emergence treatments were applied before the V3 (BBCH 13) developmental stage (imazamox), followed by cycloxydim, applied 10 days later. Herbicides were sprayed using a CO_2_ backpack sprayer (Bellspray, Inc., Opelousas, LA, USA).

### 4.3. Samples Collection

Soybean fresh biomass was measured twice: (1) 1 day before the application of the post-emergence herbicide (imazamox) (phase I) and (2) 21 days after the first post-emergence herbicide treatment (phase II). In phase I, soybean plants (five per replication) were collected from plots where pre-emergence herbicides were applied and from control plots. In phase II, plants (five per replication) were collected from all plots. Pod yield was determined at the BBCH 76 growth stage from the two central rows of each elementary plot.

Weed infestation was evaluated two weeks after post-emergence herbicide (cycloxydim) application by measuring fresh and dry biomass, weed density per square metre (above-ground weed biomass was collected from two points in each replication), and the efficacy of applied herbicides. Weed diversity was estimated using the Shannon–Wiener diversity index (H), which was calculated according to the following formula [56]:
$$H=-\sum_{i=1}^{R}pi\;ln\;pi$$
H—Shannon–Weiner diversity index; s—total number of species; pᵢ—portion of individuals belonging to species *i.*

After measuring the fresh biomass, for both soybean and weeds, samples were dried in a laboratory drying oven at 60 °C until a constant weight was achieved.

### 4.4. Statistical Analysis

The obtained data were processed by an ANOVA (F test) with a significance level of *p* < 0.05. Average values are presented with standard deviation (SD). Principal component analysis (PCA), as a dimensionality reduction method, was used to evaluate the interdependence between edamame varieties and herbicide treatments regarding weed dry biomass for each weed species, using SPSS for Windows, Version 15.0.

### 4.5. Meteorological Conditions

In 2022, the average monthly temperatures ranged from 12.6 °C in April to 25.4 °C in July, with a total of 272.5 mm of precipitation during the growing season, which, apart from June, was unevenly distributed (Figure 4). Irrigation was therefore conducted in late May (30 mm), the second decade of June (35 mm), and July (45 mm). In 2023, lower temperatures were recorded in April and May (11.7 °C and 17.8 °C, respectively), while higher temperatures occurred in July (26.4 °C) and August (24.9 °C) compared to 2022. Total precipitation was higher (347.6 mm), with notably excessive rainfall in May (~100 mm). However, a prolonged dry period in the second half of June required one irrigation of 30 mm at the beginning of the third decade of June. In 2024, except May, the average temperatures were higher compared to 2022 and 2023, while total precipitation was lower (256.6 mm). July was marked by a heat wave (maximum temperatures near 40 °C and tropical nights lasting almost 15 days) and unevenly distributed rainfall (74.1 mm mostly in two days at the beginning of the month), so irrigation was applied in the last decade of July (40 mm) (Table 3).

## 5. Conclusions

Profit-driven agriculture will expand areas being used for edamame in the future, including regions where it was not previously grown. Edamame soybean production is significantly influenced by the interaction of weather conditions, weed control, and variety choice. The results indicate that the meteorological conditions are the most dominant factor. Nevertheless, to secure edamame yield potential, weed management must take on an important role in establishing proper cropping technology, particularly in environments where edamame soybean has not been or is rarely grown, such as the Western Balkan region. The results from this research provide unique information and practical recommendations for edamame producers regarding yield potential and weed management.

Irrespective of the variety characteristics, it was shown that pre-emergence herbicide treatment could be satisfactory at reducing weed infestation when supported by favourable weather conditions, especially optimal soil humidity. Moreover, the combined application of pre- and post-emergence herbicides increases production security, mainly during seasons with higher weed infestation and unsuitable weather conditions. It should also be emphasized that Midori Giant showed more significant yield advantages under higher weed diversity, potentially related to its stronger resource competitiveness.

Due to the lack of information and research on edamame production especially in the Western Balkan region and Serbia, this research provides valuable and practical information about the cultivation and yield potential of edamame soybean, with special reference to weed management. Further research is needed to evaluate edamame cultivars and genotypes, as well as various herbicide combinations and other cultural practices, under different soil and climatic conditions, in both temperate continental and Mediterranean sub-regions of the Western Balkans, to promote the cultivation of this new dynamic crop in the wider region.

## Figures and Tables

**Figure 1 plants-14-03438-f001:**
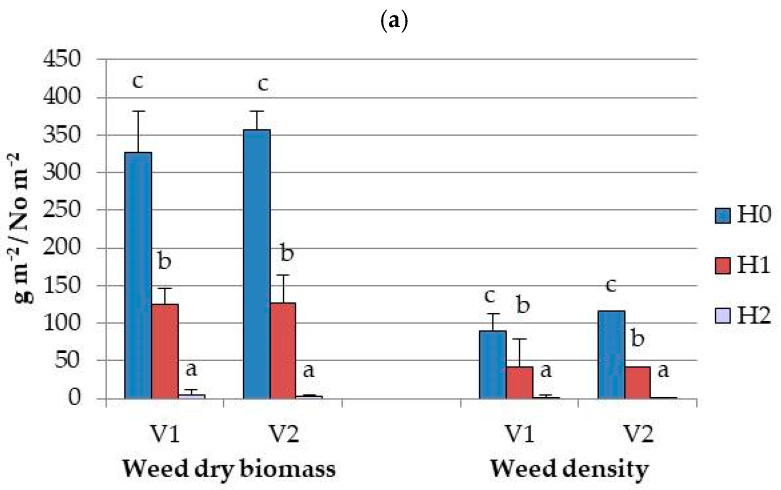
Weed dry biomass (g m^−2^) and density (plants m^−2^) (**a**), soybean dry biomass (g m^−2^) (**b**), and pod yield (t ha^−1^) (**c**) as influenced by interaction of variety (V) and herbicide treatments (H) (different letters indicate significant differences between values at *p* < 0.05; bars present standard deviation (SD) values).

**Figure 2 plants-14-03438-f002:**
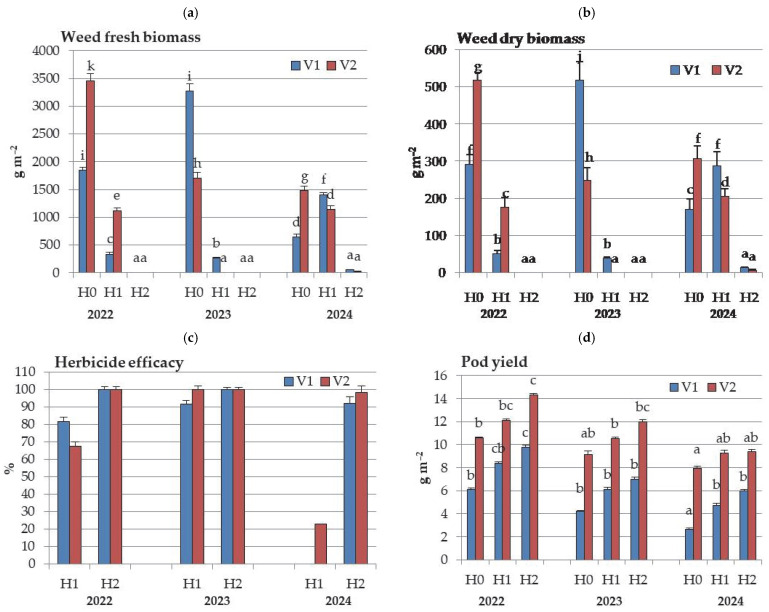
Weed fresh (**a**) and dry (**b**) biomass (g m^−2^), reduction in weed dry biomass (%) (**c**) and pod yield (t ha^−1^) (**d**) as influenced by interaction of year, variety (V), and herbicide treatments (H). (Different letters indicate significant differences between values at *p* < 0.05. Bars present standard deviation (SD) values).

**Figure 3 plants-14-03438-f003:**
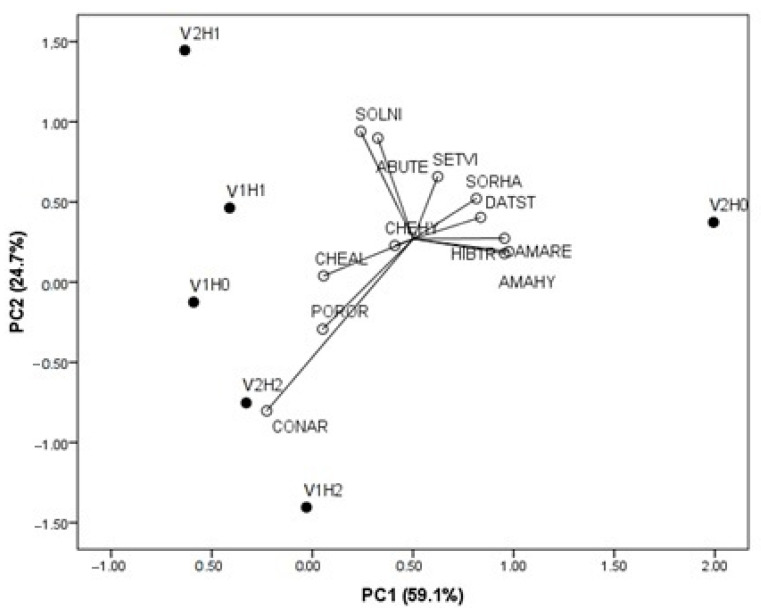
Principal component analysis (PCA) for biomass productivity of weed species in edamame crop (V1—Chiba Green; V2—Midori Giant; H0—control; H1—pre-em herbicides; H2—pre-em + post-em herbicides; SORHA—*Sorghum halepense*; CHEAL—*Chenopodium album*; AMARE—*Amaranthus retroflexus*; AMAHY—Amaranthus hybridus; SOLNI—*Solanum nigrum*; ABUTE—*Abutylon theophrasti*; DATST—*Datura stramonim*; HIBTR—*Hibiscus trionum*; CHEHY—*Chenopodium hybridum*; CONAR—*Convolvulus arvensis*; SETVI—*Setaria viridis*; POROR—*Portulaca oleracea*).

**Figure 4 plants-14-03438-f004:**
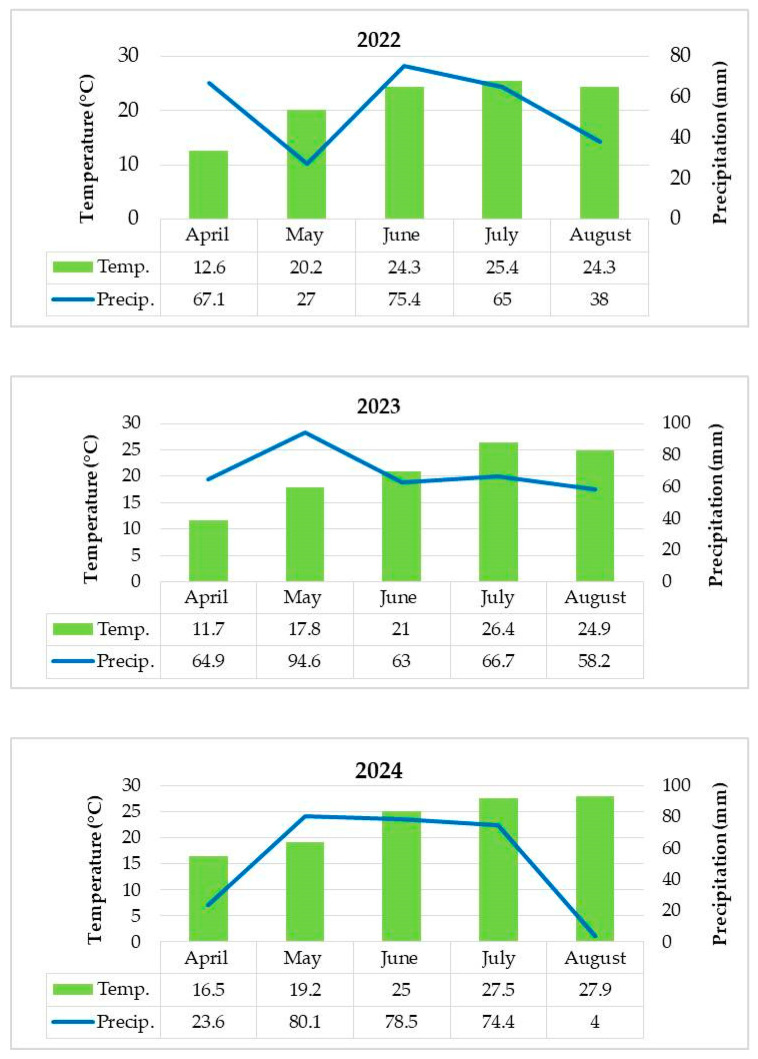
Monthly average temperature (°C) and total precipitation (mm) in 2022, 2023, and 2024 growing seasons.

**Table 1 plants-14-03438-t001:** Effects of years (Y), herbicide treatments (H), and variety (V) on fresh and dry biomass of weeds and edamame soybean and pod yield.

Year/Herbicide Treatment/ Variety	Weed FW (g m^−2^)	Weed DW (g m^−2^)	PY (t ha^−1^)	PH I *		PH II	
				SFW (g m^−2^)	SDW (g m^−2^)	SFW (g m^−2^)	SDW (g m^−2^)
**2022**	1164.3 ^n.s.^	172.2 ^n.s.^	10.2 ^b^	125.2 ^b^	23.2 ^b^	994.4 ^c^	219.3 ^c^
**2023**	873.7 ^n.s.^	134.3 ^n.s.^	8.2 ^ab^	148.6 ^c^	24.6 ^b^	558.7 ^b^	114.4 ^b^
**2024**	792.5 ^n.s.^	164.7 ^n.s.^	6.6 ^a^	71.5 ^a^	16.5 ^a^	211.2 ^a^	51.2 ^a^
**F**	0.74	0.32	11.32 *	128.54 *	9.54 *	155.54 *	145.96 *
** *p* **	0.482	0.729	0	0	0	0	0
**H0^1^**	2104.9 ^c^	341.8 ^c^	6.7 ^a^	107.5 ^n.s.^	17.7 ^n.s.^	610.7 ^n.s.^	128.5 ^n.s.^
**H1**	712.9 ^b^	126.0 ^b^	8.5 ^ab^	122.6 ^n.s.^	20.9 ^n.s.^	693.6 ^n.s.^	154.8 ^n.s.^
**H2**	12.6 ^a^	3.3 ^a^	9.7 ^b^	/	/	459.8 ^n.s.^	102.1 ^n.s.^
**F**	56.42 *	69.20 *	6.75 *	2.25	2.1	2.78	2.92
** *p* **	0	0	0.002	0.140	0.154	0.069	0.060
**V1**	869.5 ^n.s.^	151.9 ^n.s.^	6.1 ^a^	113.6 ^n.s.^	19.0 ^n.s.^	618.6 ^n.s.^	138.2 ^n.s.^
**V2**	1017.4 ^n.s.^	162.1 ^n.s.^	10.6 ^b^	116.6 ^n.s.^	19.6 ^n.s.^	557.5 ^n.s.^	118.4 ^n.s.^
**F**	0.32	0.06	93.43 *	0.08	0.08	0.51	1.17
** *p* **	0.576	0.805	0	0.774	0.779	0.472	0.283
**LSD_0.05_**							
**Y**	1116	174.2	2.645	13.94	5.56	154.1	34.44
**H**	694.7	100.9	2.787	34.88	7.772	348.0	75.52
**V**	1117	173.7	1.980	35.70	7.94	357.8	77.56
**Y * H**	476.0	70.01	2.432	11.48	3.855	70.60	20.13
**Y * V**	1075	171.7	1.322	13.33	3.884	152.9	33.00
**H * V**	685.1	102.8	1.596	35.35	7.882	353.9	76.37
**Y * H * V**	60.7	23.04	0.171	5.48	2.925	30.40	13.20

^1^ Untreated control without herbicides, H0; pre-emergence treatment, H1; pre- + post-emergence treatment, H2; Chiba Green, V1; Midori Giant, V2; phase I, PH I (1 day before post-emergence herbicide application), phase II, PH II (21 days after post-emergence herbicide application); pod yield, PY; weed fresh weight, Weed FW; weed dry weight, Weed DW; soybean fresh weight, SFW; soybean dry weight, SDW; Different letters indicate significant differences between values at *p* < 0.05; non-significant, n.s.; (*) indicates a statistically significant difference with *p* < 0.05.

**Table 2 plants-14-03438-t002:** Shannon–Wiener diversity index (H) of weed communities in two edamame varieties under different herbicide treatments over three years (2022–2024).

Shannon–Wiener Diversity Index (H)						
	V1: Chiba Green			V2: Midori Giant		
Treatment/Year	2022	2023	2024	2022	2023	2024
H0	1.22 ± 0.1	0.83 ± 0.11	1.50 ± 0.2	1.43 ± 0.14	1.60 ± 0.17	1.34 ± 0.11
H1	1.37 ± 0.1	0.77 ± 0.29	0.58 ± 0.1	1.62 ± 0.19	/	0.34 ± 0.16
H2	/ *	/	0	/	/	0.17 ± 0.17

* No species were present in the treatment.

**Table 3 plants-14-03438-t003:** Dates of applied cropping practices.

	2022	2023	2024
Sowing	21 April	3 May	23 April
Mineral fertilization	NPK—24 November 2021 UREA—29 April	NPK—4 November 2022 UREA—11 April	NPK—7 November 2023 UREA—3 April
Pre-emergence herbicide treatment	23 April	4 May	24 April
Post-emergence herbicide I treatment	19 May	24 May	23 May
Post-emergence herbicide II treatment	26 May	1 June	28 May
Biomass sampling I	18 May	24 May	23 May
Biomass sampling II and weed evaluation	7 June	13 June	10 June
Irrigation	24 May (30 mm) 21 June (35 mm) 20 July (45 mm)	20 June (50mm)	26 July (40mm)
Harvesting	V1—25 July V2—9 August	V1—9 August V2—15 August	V1—7 August V2—19 August

**Table 4 plants-14-03438-t004:** The herbicides applied in the experiment.

Trade Name	Treatment	Active Ingredient	Producer	Rate (g a.i. ha^−1^)
Dual gold 960 EC	PRE	*S*-metolachlor	Syngenta, Switzerland	1248
Lord 700 WDG	PRE	Metribuzin	Willowood, China	350
Pulsar ^®^ 40	POST	Imazamox	BASF, Germany	44
Focus ultra	POST	Cycloxydim	BASF, Germany	200

Active ingredients, a.i.

## Data Availability

The original contributions presented in this study are included in the article. Further inquiries can be directed to the corresponding author.

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
