# Peer review of "Weed Management in Edamame Soybean Production"

_plants, 2025, doi:10.3390/plants14223438_

Round 1

Reviewer 1 Report

Comments and Suggestions for Authors

The manuscript focuses on a topical issue. Appropriate evaluation and statistical methods were used. The text is understandable. I would only recommend moving the Materials and Methods section before the Results section. Below are some minor comments, recommendations, and questions.

Graphs with pod yield and weed (dry)biomass in individual growing seasons are missing.

I recommend to replace “control treatment” with “untreated control” (H0)

I recommend placing each table on one page (not two pages).

I recommend to use scientific (Latin) name of each plant (weed) species

What was the previous crop? Which herbicide was used in previous crop?

L27: replace “favourable” with “suitable”

L58: replace “broadleaf” with “dicot” or “dicotyledonous”

L63: replace “narrowleaf” with “monocot” or “monocotyledonous” or “grass”

L113: 4.5 t ha-1

L148-149: Could you explain in Discussion why was weed biomass in H1 treatment higher than in H0 treatment? I think that weeds with low competition ability was controlled by PRE herbicides and free spaces were filled by weeds with higher competitive ability.

L222: How are the critical growth phases defined? Irrigation shortly after PRE application could increase herbicide efficacy in 2024.

L250: delete “and year”

L260: replace “environments” with “weather conditions”

L292: insert “both” before “cultivars”

L302: insert “seed” before “producers”

309: replace “results” with “weight”

L364: describe growth stage V3 by BBCH

L372: Which herbicide was used 1 day before phase I?

L378: Which herbicide was used?

L380: replace “species” with “weed”

L427: replace “meteorological” with “weather”

L427: replace “sufficient rainfall” with “soil humidity”

L429: replace “greater meteorological fluctuations” with “unsuitable weather conditions”

Table 3: the column for tested varieties is missing (V1 and V2)

Table 6: the column for application term (PRE/POST) is missing. Why was used herbicide Focus? Which grasses were occurred?

Figure 2-4:  vertical axes lack units

Author Response

Open Review

Quality of English Language

( ) The English could be improved to more clearly express the research.
(x) The English is fine and does not require any improvement.

Yes

Can be improved

Must be improved

Not applicable

Does the introduction provide sufficient background and include all relevant references?

(x)

( )

( )

( )

Is the research design appropriate?

(x)

( )

( )

( )

Are the methods adequately described?

( )

(x)

( )

( )

Are the results clearly presented?

(x)

( )

( )

( )

Are the conclusions supported by the results?

(x)

( )

( )

( )

Are all figures and tables clear and well-presented?

( )

(x)

( )

( )

Comments and Suggestions for Authors

The manuscript focuses on a topical issue. Appropriate evaluation and statistical methods were used. The text is understandable. I would only recommend moving the Materials and Methods section before the Results section. Below are some minor comments, recommendations, and questions.

Reply: Thank you for support, we appreciate your valuable comments which contributed to the manuscript improvement. Regarding M&M section placement, sections are arranged according to journal template.

Graphs with pod yield and weed (dry)biomass in individual growing seasons are missing.

Reply: The results on pod yield and weed dry biomass are not referred to any Figure, but to Table 1. Nevertheless, due to your comment we decided to include into paper Figure pod yield across all three growing seasons, as a part of Figure 2.

I recommend to replace “control treatment” with “untreated control” (H0)

Reply: Thank you for the suggestion. The term has been corrected accordingly.

I recommend placing each table on one page (not two pages).

Reply: Tables have been adjusted to fit on a single page, but due to the changes in text they could be split over two pages, again.

I recommend to use scientific (Latin) name of each plant (weed) species

Reply: The Latin names of all weed species have been added throughout the manuscript.

-What was the previous crop? Which herbicide was used in previous crop?

Reply: Previous crop was wheat. Information has been added in M&M:  Previous crop was winter wheat, each year and applied herbicides were iodosulfuron-methyl-sodium + amidosulfuron and metsulfuron-methyl.

L27: replace “favourable” with “suitable”

Reply: The term has been corrected.

L58: replace “broadleaf” with “dicot” or “dicotyledonous”

Reply: The term has been corrected.

L63: replace “narrowleaf” with “monocot” or “monocotyledonous” or “grass”

Reply: The term has been corrected.

L113: 4.5 t ha-1

Reply: The unit has been completed as 4.5 t ha⁻¹.

L148-149: Could you explain in Discussion why was weed biomass in H1 treatment higher than in H0 treatment? I think that weeds with low competition ability was controlled by PRE herbicides and free spaces were filled by weeds with higher competitive ability.

Reply: The explanation has been added in Discussion accordingly: Besides, the pre-emergence herbicide in the H1 treatment likely suppressed less competitive weed species, creating more space and enabling the late-emerging weed species, which are more tolerant, with greater biomass productivity to occupy the free space [39].What is more, the application of pre-emergence herbicides can cause shifts in weed flora characteristics as reflected by changes in species composition and diversity [40].

L222: How are the critical growth phases defined? Irrigation shortly after PRE application could increase herbicide efficacy in 2024.

Reply: The critical growth stages of soybean, in terms of water demand, are generally defined as the reproductive phases, from flowering to pod filling when water deficit has the greatest impact on yield formation. We agree that irrigation shortly after PRE application could increase herbicide efficacy in 2024, but edamame sowing was conducted when soil moisture was optimal, and pre-em treatment was applied 1 day after sowing, so we considered that soil moisture was optimal for herbicide activation. We were mainly focused on irrigation during soybean growing period, as the critical.

L250: delete “and year”

Reply: It has been deleted as suggested.

L260: replace “environments” with “weather conditions”

Reply: It has been replaced as suggested.

L292: insert “both” before “cultivars”

Reply: “both” has been inserted.

L302: insert “seed” before “producers”

Reply: it was inserted.

309: replace “results” with “weight”

Reply: it has been replaced.

L364: describe growth stage V3 by BBCH

Reply: Thank you for the suggestion. The V3 growth stage has been described following the BBCH scale.

L372: Which herbicide was used 1 day before phase I?

Reply: The herbicide applied 1 day before Phase I was Imazamox , and this information has now been indicated.

L378: Which herbicide was used?

Reply: herbicide cycloxydim was used, and this information has now been indicated

L380: replace “species” with “weed”

Reply: it has been replaced as suggested.

L427: replace “meteorological” with “weather”

Reply: it has been replaced as suggested.

L427: replace “sufficient rainfall” with “soil humidity”

Reply: it has been replaced as suggested.

L429: replace “greater meteorological fluctuations” with “unsuitable weather conditions”

Reply: it has been replaced as suggested.

Table 3: the column for tested varieties is missing (V1 and V2)

Reply: There are rows which explained results for each variety (V1: Chiba green, V2: Midori giant is included prior to the results. Nevertheless, table 3 was replaced with Figure 2 and we hope that results are now presented clearly.

Table 6: the column for application term (PRE/POST) is missing. Why was used herbicide Focus? Which grasses were occurred?

Reply: The column has been added. Grasses occured in experimantal plot were Sorghum halepense and Setaria viridis.

Figure 2-4:  vertical axes lack units

Reply: Units were added to the vertical axes.

Reviewer 2 Report

Comments and Suggestions for Authors

In manuscript plants-3938413, the authors evaluated two herbicide management programs in edamame soybean production over three growing seasons. The study presents interesting findings; however, although the main factors influencing the results were the environmental conditions of each season and the specific soybean variety, the description of the results and the discussion sections failed to include the herbicides used.

Moreover, the section that could be most improved is the Results section. First, the writing could be revised to avoid subjective expressions; second, instead of presenting results only in tables, it would be preferable to use composite figures (with subfigures for each parameter). This would give the manuscript a more impactful visual presentation, make data interpretation easier, and likely attract more readers—thus increasing its citation potential.

Below are some specific comments:

  • L13–32: Provide a detailed description of the herbicide treatments (PRE and PRE + POST).
  • L21, 24: Delete “significantly.”
  • L38: Correct to “Serbia WAS…”.
  • L45–51: Add appropriate references to support these claims.
  • L54: Specify that soybean is a C3 plant.
  • L68: Clarify whether these statements are general or specific to soybean.
  • L87: Begin a new paragraph here.
  • L100: Indicate numerically which treatment achieved the highest yield.
  • L98–105: To allow relative comparisons, include at least one reference value. Although the text indicates how much a variable increased or decreased, readers cannot gauge the magnitude of the change without a baseline. For example, if parameter B was three times (300%) higher than A, it would be important to report the mean of A (at least in parentheses). Add key reference values for L102 (biomass accumulation) and L104 (fresh biomass).
  • L106, 112, 113, 122, 125, 144, 149: Delete “significantly/significant.” This expression is subjective and redundant since any statement about results should already be statistically supported. Avoid other subjective expressions (e.g., superlatives ending in -est) in the results description, and instead, report numerical values directly. Adjust all results as needed.
  • L108: Provide numerical values for mass.
  • L109: Report numerically (absolute or relative) the greatest biomass reduction.
  • L117: Convert these results into graphs, including standard error or standard deviation bars, and indicate the sample size (n). Purely statistical parameters can remain in tables and be presented as supplementary material.
  • L134, L155: Convert these results into figures.
  • L182–187: Use only one decimal place for percentage values.
  • L189: Delete “According to Figure 1,” and mention the figure only in parentheses at the end of the statement (L191).
  • L362: Specify which herbicides were applied in each treatment. For the PRE treatments, were S-metolachlor and metribuzin applied in mixture?
  • L362: H1: Pre-emergence herbicides (S-metolachlor + metribuzin); H2: combination of pre-emergence (S-metolachlor + metribuzin) and post-emergence herbicides (imazamox and cycloxidim); H0: control (without herbicides) (Table 6).
  • L364: Although this is mentioned in the table, it would also be important to specify here that pre-emergence herbicide application was carried out 1–2 days after soybean sowing.
  • L95–341: Although it is understood that this was not the main focus of the study, it would be important to mention the herbicides both in the Results and Discussion sections, emphasizing their mechanisms of action and the basis of their selectivity in soybean. Metribuzin and S-metolachlor show positional selectivity, cycloxidim acts selectively by mechanism (affecting only grasses), and imazamox is selective in soybean due to metabolic degradation. However, some cultivars are tolerant to imidazolinones (Clearfield). The manuscript does not specify whether the soybean varieties used in this study belong to this group (Clearfield) or not.

Author Response

Open Review

(x) I would not like to sign my review report

( ) I would like to sign my review report

Quality of English Language

( ) The English could be improved to more clearly express the research.

(x) The English is fine and does not require any improvement.

Yes      Can be improved        Must be improved      Not applicable

Does the introduction provide sufficient background and include all relevant references?

(x)       ( )        ( )        ( )

Is the research design appropriate?

(x)       ( )        ( )        ( )

Are the methods adequately described?

( ) (x)   ( )        ( )

Are the results clearly presented?

( ) ( )    (x)       ( )

Are the conclusions supported by the results?

(x)       ( )        ( )        ( )

Are all figures and tables clear and well-presented?

( ) ( )    (x)       ( )

Comments and Suggestions for Authors

In manuscript plants-3938413, the authors evaluated two herbicide management programs in edamame soybean production over three growing seasons. The study presents interesting findings; however, although the main factors influencing the results were the environmental conditions of each season and the specific soybean variety, the description of the results and the discussion sections failed to include the herbicides used.

Moreover, the section that could be most improved is the Results section. First, the writing could be revised to avoid subjective expressions; second, instead of presenting results only in tables, it would be preferable to use composite figures (with subfigures for each parameter). This would give the manuscript a more impactful visual presentation, make data interpretation easier, and likely attract more readers—thus increasing its citation potential.

Reply: Thank you for the valuable comments, they surely contributed to the manuscript quality.

Below are some specific comments:

  1. L13–32: Provide a detailed description of the herbicide treatments (PRE and PRE + POST).

Reply: Thank you for your suggestion. Detailed description of herbicide treatments has been provided: Treatments included: pre-emergence herbicides - (S-metolachlor + metribuzin) (H1), pre- (S-metolachlor + metribuzin) + post-emergence herbicides (imazamox + cycloxydim) (H2), and the untreated control, (H0).

  1. L21, 24: Delete “significantly.”

Reply: It has been deleted.

  1. L38: Correct to “Serbia WAS…”.

Reply: it has been corrected.

  1. L45–51: Add appropriate references to support these claims.

Reply: references has been added as suggested.

  1. L54: Specify that soybean is a C3 plant.

Reply: The sentence was modified accordingly: Given that soybean is a C3 plant, it has lower photosynthetic efficiency and slower early growth rate than many C4 weed species, thus special attention should be given to weed management.

  1. L68: Clarify whether these statements are general or specific to soybean.

Reply: Thank you for the suggestion. These statements regarding weeds are specific for soybean, which has been clarified as you suggested.

  1. L87: Begin a new paragraph here.

Reply: Thank you for the notice, it was done.

  1. L100: Indicate numerically which treatment achieved the highest yield.

Reply: The sentence was added: Accordingly, the highest average yield was achieved at H2 treatment (9.7 t ha-1).

  1. L98–105: To allow relative comparisons, include at least one reference value. Although the text indicates how much a variable increased or decreased, readers cannot gauge the magnitude of the change without a baseline. For example, if parameter B was three times (300%) higher than A, it would be important to report the mean of A (at least in parentheses). Add key reference values for L102 (biomass accumulation) and L104 (fresh biomass).

Reply: The 1st paragraph of 2.1. chapter was modified accordingly: The growing season did not influence weed fresh and dry biomass, while it affected pod yield, as well as fresh and dry soybean biomass in both evaluated: phase I and phase II. The greatest average pod yield was achieved in 2022 (10.2 t ha⁻1), while in 2023 and 2024 it was reduced by 36% (8.2 t ha⁻1) and 50.4% (6.6 t ha⁻1), respectively. In 2022, edamame exhibited the highest average biomass accumulation, with a 7.9-fold increase in fresh mass (from 125.2 g m⁻2 in phase I to 994.4 g m⁻2 in phase II) and a 9.5-fold increase in dry mass in phase II compared to phase I (from 23.2 g m⁻2 to 219.3 g m⁻2) whereas, in 2023 the increases were 3.8-fold for fresh biomass (from 148.6 g m⁻2 in phase I to 558.7 g m⁻2 in phase II) and 4.7-fold for dry biomass(from 24.6 g m⁻2 in phase I to 114.4 g m⁻2 phase II); in 2024 increases were 3.0-fold (from 75.1 g m⁻2 in phase I to 221.2 g m⁻2 in phase II) and 5.1-fold (from 16.5 g m⁻2 in phase I to 51.2 g m⁻2 in phase II), respectively for fresh and dry biomass (Table 1).

  1. L106, 112, 113, 122, 125, 144, 149: Delete “significantly/significant.” This expression is subjective and redundant since any statement about results should already be statistically supported. Avoid other subjective expressions (e.g., superlatives ending in -est) in the results description, and instead, report numerical values directly. Adjust all results as needed.

Reply: We made changes across the results, as it was suggested. In some cases, we used the terms the greatest or the highest to underline value, regarding the other values, but we also included numerical values, as a support.

  1. L108: Provide numerical values for mass.

Reply: Numerical values were provided accordingly: Herbicide treatments affected weed biomass and pod yield. Weed dry mass averaged 3.3 g m⁻² in pre- + post-emergence treatment (H2), 126 g m⁻² in pre-emergence treatment (H1), and 341.8 g m⁻² in the untreated control (H0). The highest weed fresh and dry biomass was recorded in control plots (2104.6 g m-2 and 341.8 g m-2, respectively), while the greatest weed biomass reduction occurred in H2 treatment (167 fold for fresh weed biomass and 104 fold for weed dry biomass) regarding to H0.

  1. L109: Report numerically (absolute or relative) the greatest biomass reduction.

Reply: It was reported (answered upper, in comment 11).

  1. L117: Convert these results into graphs, including standard error or standard deviation bars, and indicate the sample size (n). Purely statistical parameters can remain in tables and be presented as supplementary material.

Reply: Thank you for the suggestion, but we consider that general ANOVA for tested sources of variability could be present in the best way in the form of table. All other tables were modified and presented as Figures, including SD values, as you indicated. Sample size and sampling procedures were described in M&M section.

  1. L134, L155: Convert these results into figures.

Reply: Both tables were converted into figures.

  1. L182–187: Use only one decimal place for percentage values.

Reply: This has been changed.

  1. L189: Delete “According to Figure 1,” and mention the figure only in parentheses at the end of the statement (L191).

Reply: this has been changed.

  1. L362: Specify which herbicides were applied in each treatment. For the PRE treatments, were S-metolachlor and metribuzin applied in mixture?

Reply: Herbicides in treatments have been specified

  1. L362: H1: Pre-emergence herbicides (S-metolachlor + metribuzin); H2: combination of pre-emergence (S-metolachlor + metribuzin) and post-emergence herbicides (imazamox and cycloxidim); H0: control (without herbicides) (Table 6).

Reply: Sentence was changed, accordingly.

  1. L364: Although this is mentioned in the table, it would also be important to specify here that pre-emergence herbicide application was carried out 1–2 days after soybean sowing.

Reply: The sentence was changed, it has been specified.

  1. L95–341: Although it is understood that this was not the main focus of the study, it would be important to mention the herbicides both in the Results and Discussion sections, emphasizing their mechanisms of action and the basis of their selectivity in soybean. Metribuzin and S-metolachlor show positional selectivity, cycloxidim acts selectively by mechanism (affecting only grasses), and imazamox is selective in soybean due to metabolic degradation. However, some cultivars are tolerant to imidazolinones (Clearfield). The manuscript does not specify whether the soybean varieties used in this study belong to this group (Clearfield) or not.

Reply: Thank you for the comment. We do agree this was not the main focus of the study.

However we provided a short paragraph explaining the nature of herbicide selectivity in soybean. Furthermore, in the M&M we added – selected varieties originated from organic production (certified by the seller) and are not genetically modified or have any herbicide trait delivered through the breeding process (e.g. herbicide tolerant cultivars).

Reviewer 3 Report

Comments and Suggestions for Authors
  1. Abstract: Optimize accuracy of expression: Change "promoted weed diversity goes in the line with yield increase" to "weed diversity was associated with yield increase in Midori Giant" to clarify the varietal specificity of this conclusion and avoid overgeneralization.
  2. Abstract should highlight the core mechanism: Add the key reason why combined herbicide (H2) application improves yield stability (e.g., "by extending the weed control period through pre- + post-emergence herbicide combinations, mitigating the impact of meteorological fluctuations on herbicide efficacy").
  3. Keywords: "herbicides" and "herbicide efficacy" have semantic overlap. It is recommended to delete "herbicides" and add "meteorological conditions" or "variety competitiveness" to better reflect the study variables.
  4. Clarify the difference in weed competition between edamame and conventional soybean: The introduction only mentions that edamame has an "earlier harvest time and lower planting density". It should be supplemented with "whether the weaker early competitiveness of edamame leads to more severe weed problems" to strengthen the research rationale.
  5. Comprehensiveness of literature review: Supplement the literature on weed management in edamame in similar climatic zones (e.g., Mediterranean region) within the Western Balkans, comparing the regional uniqueness of this study. If none exist, clearly state "this study is the first report in this region".
  6. The original objective was to "evaluate the effects of herbicide treatment, growing season, and cultivar on weed control and crop productivity". It is recommended to add "elucidate the physiological mechanisms by which meteorological conditions regulate herbicide efficacy" to enhance the research depth.
  7. In Table 1, "PH I" and "PH II" must be clearly defined in the notes as "Phase I (1 day before post-emergence herbicide application)" and "Phase II (21 days after post-emergence herbicide application)" to avoid ambiguity.
  8. Discussion: The original text mentions "yield increase in Midori Giant coincided with promoted weed diversity". This should be combined with varietal characteristics (e.g., "Midori Giant's greater plant height and denser canopy might suppress dominant weeds through resource competition, promoting the coexistence of minor weeds and reducing the competitive pressure from individual weed species"). Cite studies like Place et al. (2011) on the relationship between crop canopy and weed diversity to support this mechanism.
  9. The conclusion stating "provides recommendations for the Western Balkan region" should be cautious, as the study was conducted only in Serbia. It is recommended to add "differences between the Serbian climate (temperate continental) and the coastal areas of the Western Balkans (Mediterranean climate), suggesting the need for future validation across different sub-regions".
  10. Consistency between conclusions and results: The phrase "promoted weed diversity goes in line with yield increase" should be qualified as "Midori Giant showed more significant yield advantages under higher weed diversity, potentially related to its stronger resource competitiveness" to avoid overgeneralization.

Author Response

Open Review

Quality of English Language

(x) The English could be improved to more clearly express the research.
( ) The English is fine and does not require any improvement.

Yes

Can be improved

Must be improved

Not applicable

Does the introduction provide sufficient background and include all relevant references?

( )

( )

(x)

( )

Is the research design appropriate?

( )

( )

(x)

( )

Are the methods adequately described?

( )

( )

(x)

( )

Are the results clearly presented?

( )

( )

(x)

( )

Are the conclusions supported by the results?

( )

( )

(x)

( )

Are all figures and tables clear and well-presented?

( )

( )

(x)

( )

Comments and Suggestions for Authors

  1. Abstract: Optimize accuracy of expression: Change "promoted weed diversity goes in the line with yield increase" to "weed diversity was associated with yield increase in Midori Giant" to clarify the varietal specificity of this conclusion and avoid overgeneralization.

Reply: Thank you for the suggestion. The Abstract has been updated to clarify the varietal specificity:  Besides, weed diversity was associated with yield increase in Midori Giant.

  1. Abstract should highlight the core mechanism: Add the key reason why combined herbicide (H2) application improves yield stability (e.g., "by extending the weed control period through pre- + post-emergence herbicide combinations, mitigating the impact of meteorological fluctuations on herbicide efficacy").

Reply: Thank you for the comment, it was very useful and sentence was changed accordingly: while combined application of pre- and post-emergence herbicides increases security of production (particularly in the seasons with higher weed infestation), likely by extending the weed control period through pre- + post-emergence herbicide combinations, targeting different weed species over soybean vegetative period.

  1. Keywords: "herbicides" and "herbicide efficacy" have semantic overlap. It is recommended to delete "herbicides" and add "meteorological conditions" or "variety competitiveness" to better reflect the study variables.

Reply: Key words were changed: vegetable soybean; herbicide efficacy; pod yield; variety competitiveness; meteorological conditions; weed biomass; weed control;

  1. Clarify the difference in weed competition between edamame and conventional soybean: The introduction only mentions that edamame has an "earlier harvest time and lower planting density". It should be supplemented with "whether the weaker early competitiveness of edamame leads to more severe weed problems" to strengthen the research rationale.

Reply: Thank you for the comment, the last paragraph of Introduction was modified, accordingly: Although no studies in the region, have specifically compared the early-season competitive ability of edamame and conventional soybean, it is generally recognized that crops established at lower planting densities, develop the canopy more slowly and are therefore less competitive with early-emerging, aggressive weeds [26]. Combined with the limited number of registered herbicides for this crop, these traits can lead to more severe weed problems if not properly managed [27-28].

  1. Comprehensiveness of literature review: Supplement the literature on weed management in edamame in similar climatic zones (e.g., Mediterranean region) within the Western Balkans, comparing the regional uniqueness of this study. If none exist, clearly state "this study is the first report in this region".

Reply: as there were no studies about weed management in edamame, your suggestion to emphasise that was added in introduction part: Therefore, this research is the first report about weed management in edamame in this region.

  1. The original objective was to "evaluate the effects of herbicide treatment, growing season, and cultivar on weed control and crop productivity". It is recommended to add "elucidate the physiological mechanisms by which meteorological conditions regulate herbicide efficacy" to enhance the research depth.

Reply: Thank you for the valuable comment; we included few sentences in Discussion regarding meteorological conditions on herbicide effectiveness: While the higher temperatures increase herbicide absorption and translocation, they also increase their further metabolisation, thus reducing the efficacy [38], with potential incidence of phytotoxic effects on crops. The soil conditions are also the important factor when herbicide efficacy was considered, where the higher soil temperature increases the rate of herbicide degradation, due to the increased microbial activity and chemical degradation. Besides, the pre-emergence herbicide in the H1 treatment likely suppressed less competitive weed species, creating more space and enabling the late-emerging weed species, which are more tolerant, with greater biomass productivity to occupy the free space [39].What is more, the application of pre-emergence herbicides can cause shifts in weed flora characteristics as reflected by changes in species composition and diversity [40].At the same time the lowest weeds biomass was recorded in 2024 comparing to the other experimental years. Taking into account greater weeds adaptability potential, the harsh meteorological conditions, herbicide effectiveness was decreased. Meanwhile, in 2023, pre-emergence herbicide application proved success, suggesting that weather conditions, particularly sufficient precipitation amount following the herbicide application, enhanced herbicide activation and efficacy, so the post-emergence treatment could be omitted. This aligns with findings by Meyer (2023) [41], who stated that the effectiveness of pre-emergence herbicides is highly dependent on soil moisture availability after the treatment, in order to transfer the herbicide into the weeds seed zone and enhance herbicide uptake by roots [38]. It should be also mentioned that herbicide efficacy depends also on weed water status, where water-stressed weeds will poorly absorb and translocate systemic herbicides.

  1. In Table 1, "PH I" and "PH II" must be clearly defined in the notes as "Phase I (1 day before post-emergence herbicide application)" and "Phase II (21 days after post-emergence herbicide application)" to avoid ambiguity.

Reply: Thank you for the comment. The notes in Table 1 have been revised to clearly define PH I and PH II as suggested.

  1. Discussion: The original text mentions "yield increase in Midori Giant coincided with promoted weed diversity". This should be combined with varietal characteristics (e.g., "Midori Giant's greater plant height and denser canopy might suppress dominant weeds through resource competition, promoting the coexistence of minor weeds and reducing the competitive pressure from individual weed species"). Cite studies like Place et al. (2011) on the relationship between crop canopy and weed diversity to support this mechanism.

Reply: Thank you for the valuable comment, we modified this part of the text accordingly: Taking into account greater weed biomass productivity in 2022 and 2024 (as opposite years from the meteorological viewpoint), together with the greater weeds diversity, and greater pod yield potential of V2, it could be supposed that Midori Giant has greater height, with potentially denser canopy [13, 31], what could suppress dominant weeds through resource competition. Consequently, it could support V2 capacity to increase yield when the greater weed diversification was present, supporting its overall competitiveness [45] in regard to Chiba Green.

Also, there is more detailed explanation at the end of 3.1 section: Considering variety, it influenced only pod yield, where Midori Giant, as variety with longer vegetative period and greater biomass production had significantly higher yield than Chiba Green. Williams [27] compared edamame cultivars from different maturity groups and plant canopy characteristics, and noted that the late-maturing cultivar, with greater height and denser canopy also had a higher yield. It is also obvious that there is no difference in pod yield between H1 and H2 treatment, for both varieties. Although Midori Giant (V2) and Chiba Green (V1) are classified within the same maturity group [13, 31] our observations indicate that V2 required approximately 10 additional days to reach the BBCH 76 stage (full seed development) than V1. According to Jankauskienė et al., [13], no differences were observed between Midori Giant and Chiba Green in pod yield. Both studies [13, 31] also reported that Midori Giant is taller than Chiba Green, what may have contributed to greater weed competitiveness of Midori, comparing to Chiba. On the other hand, Ogles et al., [31] described Chiba Green as the higher-yielding variety, indicating that yield performance may vary depending on environmental conditions.

  1. The conclusion stating "provides recommendations for the Western Balkan region" should be cautious, as the study was conducted only in Serbia. It is recommended to add "differences between the Serbian climate (temperate continental) and the coastal areas of the Western Balkans (Mediterranean climate), suggesting the need for future validation across different sub-regions".

Reply: As you suggested it was added: Further research is needed to evaluate edamame cultivars and genotypes, as well as various herbicide combinations and other cultural practices, under different soil and climatic conditions, both in temperate continental and Mediterranean sub-regions of Western Balkans to promote the cultivation of this new dynamic crop in the wider region.

  1. Consistency between conclusions and results: The phrase "promoted weed diversity goes in line with yield increase" should be qualified as "Midori Giant showed more significant yield advantages under higher weed diversity, potentially related to its stronger resource competitiveness" to avoid overgeneralization.

Reply: Suggested sentence has been replaced.

Round 2

Reviewer 2 Report

Comments and Suggestions for Authors

The authors have addressed all my comments, adjusted the description of the results, and converted several tables into figures. The document contains multiple typographical errors, mainly double spaces and, in some cases, missing spaces. However, intellectually, the manuscript is acceptable for publication, as these details will be corrected during proofreading.
L76: delete the space to join the paragraphs.
L390: each 5 m long.
L454: (30 mm) (Table 3).

Author Response

Comments and Suggestions for Authors

The authors have addressed all my comments, adjusted the description of the results, and converted several tables into figures. The document contains multiple typographical errors, mainly double spaces and, in some cases, missing spaces. However, intellectually, the manuscript is acceptable for publication, as these details will be corrected during proofreading.

Response: Thank you for all suggestions, they significantly contributed to the manuscript quality. Due to the multiple editing processes, including the language checking and correcting, some errors may appear in the text. We tried to correct all typographical errors in the text.

L76: delete the space to join the paragraphs.

Response: Space was deleted and paragraphs were joined.

L390: each 5 m long.

Response: Sorry for misunderstanding, there was no pointed part of the text at L390. But, it seems that pointed part is referred to 4.1. Section, so we corrected sentence, accordingly: The size of elementary plot was 20 m2, consisting of six soybean rows each 5 m long.

L454: (30 mm) (Table 3).

Response: Thank you for the notice, data in Table 3 was corrected, as well as in text following meteorological conditions (Chapter 4.5) for 2024.

We also noticed some errors in Figures, which were corrected, accordingly.

Reviewer 3 Report

Comments and Suggestions for Authors

The author has made good revisions to the manuscript.

Author Response

Comments and Suggestions for Authors

The author has made good revisions to the manuscript.

Response: Thank you your effort and suggestion which contributed to the manuscript quality.

We checked the manuscript again and included corrections of text, Figures, and language.